# Answering Complex Open-Domain Questions with Multi-Hop Dense Retrieval

**Wenhan Xiong**[1*]  **Xiang Lorraine Li**[2*]  **Srinivasan Iyer**[‡]  **Jingfei Du**[‡]

**Patrick Lewis**[‡†]  **William Wang**[1]  **Yashar Mehdad**[‡]  **Wen-tau Yih**[‡]

**Sebastian Riedel**[‡†]  **Douwe Kiela**[‡]  **Barlas Oğuz**[‡]

[1]University of California, Santa Barbara
[2]University of Massachusetts Amherst
[‡]Facebook AI
[†]University College London
{xwhan, william}@cs.ucsb.edu, xiangl@cs.umass.edu,
{sviyer, jingfeidu, plewis, mehdad, scottyih, sriedel, dkiela, barlaso}@fb.com

## Abstract

We propose a simple and efficient multi-hop dense retrieval approach for answering complex open-domain questions, which achieves state-of-the-art performance on two multi-hop datasets, HotpotQA and multi-evidence FEVER. Contrary to previous work, our method does not require access to any corpus-specific information, such as inter-document hyperlinks or human-annotated entity markers, and can be applied to any unstructured text corpus. Our system also yields a much better efficiency-accuracy trade-off, matching the best published accuracy on HotpotQA while being 10 times faster at inference time.[1]

## 1 Introduction

*Open domain question answering* is a challenging task where the answer to a given question needs to be extracted from a large pool of documents. The prevailing approach (Chen et al., 2017) tackles the problem in two stages. Given a question, a *retriever* first produces a list of $k$ candidate documents, and a *reader* then extracts the answer from this set. Until recently, retrieval models were dependent on traditional term-based information retrieval (IR) methods, which fail to capture the semantics of the question beyond lexical matching and remain a major performance bottleneck for the task. Recent work on dense retrieval methods instead uses pretrained encoders to cast the question and documents into dense representations in a vector space and relies on fast maximum inner-product search (MIPS) to complete the retrieval. These approaches (Lee et al., 2019; Guu et al., 2020; Karpukhin et al., 2020) have demonstrated significant retrieval improvements over traditional IR baselines.

However, such methods remain limited to *simple* questions, where the answer to the question is explicit in a single piece of text evidence. In contrast, *complex* questions typically involve aggregating information from multiple documents, requiring logical reasoning or sequential (multi-hop) processing in order to infer the answer (see Figure 1 for an example). Since the process for answering such questions might be sequential in nature, single-shot approaches to retrieval are insufficient. Instead, iterative methods are needed to recursively retrieve new information at each step, conditioned on the information already at hand. Beyond further expanding the scope of existing textual open-domain QA systems, answering more complex questions usually involves *multi-hop reasoning*, which poses unique challenges for existing neural-based AI systems. With its practical

---

[*]Equal Contribution
[1]https://github.com/facebookresearch/multihop_dense_retrieval.

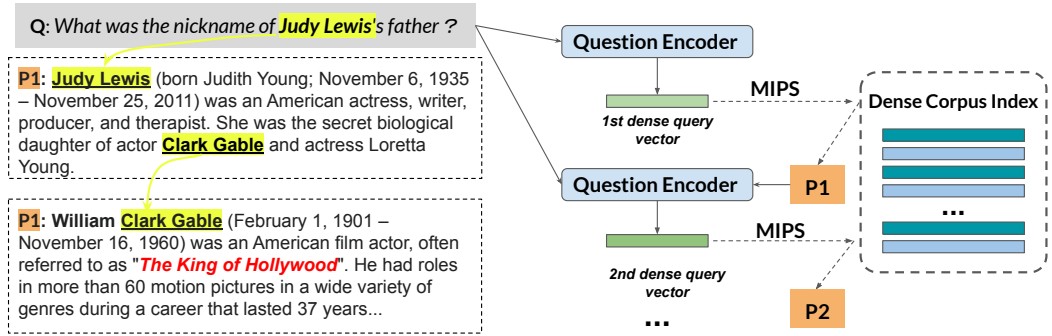

Figure 1: An overview of the multi-hop dense retrieval approach.

and research values, multi-hop QA has been extensively studied recently (Talmor & Berant, 2018; Yang et al., 2018; Welbl et al., 2018) and remains an active research area in NLP (Qi et al., 2019; Nie et al., 2019; Min et al., 2019; Zhao et al., 2020; Asai et al., 2020; Perez et al., 2020).

The main problem in answering multi-hop open-domain questions is that the search space grows exponentially with each retrieval hop. Most recent work tackles this issue by constructing a document graph utilizing either entity linking or existing hyperlink structure in the underlying Wikipedia corpus (Nie et al., 2019; Asai et al., 2020). The problem then becomes finding the best path in this graph, where the search space is bounded by the number of hyperlinks in each passage. However, such methods may not generalize to new domains, where entity linking might perform poorly, or where hyperlinks might not be as abundant as in Wikipedia. Moreover, efficiency remains a challenge despite using these data-dependent pruning heuristics, with the best model (Asai et al., 2020) needing hundreds of calls to large pretrained models to produce a single answer.

In contrast, we propose to employ dense retrieval to the multi-hop setting with a simple recursive framework. Our method iteratively encodes the question and previously retrieved documents as a query vector and retrieves the next relevant documents using efficient MIPS methods. With high-quality, dense representations derived from strong pretrained encoders, our work first demonstrates that the sequence of documents that provide sufficient information to answer the multi-hop question can be accurately discovered from unstructured text, *without* the help of corpus-specific hyperlinks. When evaluated on two multi-hop benchmarks, HotpotQA (Yang et al., 2018) and a multi-evidence subset of FEVER (Thorne et al., 2018), our approach improves greatly over the traditional linking-based retrieval methods. More importantly, the better retrieval results also lead to state-of-the-art downstream results on both datasets. On HotpotQA, we demonstrate a vastly improved efficiency-accuracy trade-off achieved by our system: by limiting the amount of retrieved contexts fed into downstream models, our system can match the best published result while being 10x faster.

## 2 METHOD

### 2.1 PROBLEM DEFINITION

The retrieval task considered in this work can be described as follows (see also Figure 1). Given a multi-hop question $q$ and a large text corpus $\mathcal{C}$, the retrieval module needs to retrieve a sequence of passages $\mathcal{P}_{seq} : \{p_1, p_2, ..., p_n\}$ that provide *sufficient* information for answering $q$. Practically, the retriever returns the $k$ best-scoring sequence candidates, $\{\mathcal{P}_{seq}^1, \mathcal{P}_{seq}^2, ..., \mathcal{P}_{seq}^k\}$ ($k \ll |\mathcal{C}|$), with the hope that at least one of them has the desired qualities. $k$ should be small enough for downstream modules to process in a reasonable time while maintaining adequate recall. In general, retrieval also needs to be efficient enough to handle real-world corpora containing millions of documents.

### 2.2 MULTI-HOP DENSE RETRIEVAL

**Model**    Based on the sequential nature of the multi-hop retrieval problem, our system solves it in an iterative fashion. We model the probability of selecting a certain passage sequence as follows:

$$P(\mathcal{P}_{seq}|q) = \prod_{t=1}^{n} P(p_t|q, p_1, ..., p_{t-1}),$$

where for $t = 1$, we only condition on the original question for retrieval. At each retrieval step, we construct a new query representation based on previous results and the retrieval is implemented as maximum inner product search over the dense representations of the whole corpus:

$$P(p_t|q, p_1, ..., p_{t-1}) = \frac{\exp\left(\langle \boldsymbol{p}_t, \boldsymbol{q}_t \rangle\right)}{\sum_{p \in \mathcal{C}} \exp\left(\langle \boldsymbol{p}, \boldsymbol{q}_t \rangle\right)}, \text{ where } \boldsymbol{q}_t = g(q, p_1, ..., p_{t-1}) \text{ and } \boldsymbol{p}_t = h(p_t).$$

Here $\langle \cdot, \cdot \rangle$ is the inner product between the query and passage vectors. $h(\cdot)$ and and $g(\cdot)$ are passage and query encoders that produce the dense representations. In order to reformulate the query representation to account for previous retrieval results at time step $t$, we simply concatenate the question and the retrieved passages as the inputs to $g(\cdot)$. Note that our formulation for each retrieval step is similar to existing single-hop dense retrieval methods (Lee et al., 2019; Guu et al., 2020; Karpukhin et al., 2020) except that we add the query reformulation process conditioned on previous retrieval results. Additionally, instead of using a bi-encoder architecture with separately parameterized encoders for queries and passages, we use a shared RoBERTa-base (Liu et al., 2019) encoder for both $h(\cdot)$ and $g(\cdot)$. In §3.1.3, we show this simple modification yields considerable improvements. Specifically, we apply layer normalization over the start token's representations from RoBERTa to get the final dense query/passage vectors.

**Training and Inference**    The retriever model is trained as in Karpukhin et al. (2020), where each input query (which at each step consists of a question and previously retrieved passages) is paired with a positive passage and $m$ negative passages to approximate the softmax over all passages. The positive passage is the gold annotated evidence at step $t$. Negative passages are a combination of passages in the current batch which correspond to other questions (*in-batch*), and *hard* negatives which are false adversarial passages. In our experiments, we obtain hard negatives from TF-IDF retrieved passages and their linked pages in Wikipedia. We note that using hyperlinked pages as additional negatives is neither necessary nor critical for our approach. In fact we observe only a very small degradation in performance if we remove them from training (§3.1.3). In addition to in-batch negatives, we use a memory bank ($\mathcal{M}$) mechanism (Wu et al., 2018) to further increase the number of negative examples for each question. The memory bank stores a large number of dense passage vectors. As we block the gradient back-propagation in the memory bank, its size ($|\mathcal{M}| \gg$ batch size) is less restricted by the GPU memory size. Specifically, after training to convergence with the shared encoder, we freeze a copy of the encoder as the new passage encoder and collect a bank of passage representations across multiple batches to serve as the set of negative passages. This simple extension results in further improvement in retrieval. (§3.1.3).

For inference, we first encode the whole corpus into an index of passage vectors. Given a question, we use beam search to obtain top-$k$ passage sequence candidates, where the candidates to beam search at each step are generated by MIPS using the query encoder at step $t$, and the beams are scored by the sum of inner products as suggested by the probabilistic formulation discussed above. Such inference relies only on the dense passage index and the query representations, and does not need explicit graph construction using hyperlinks or entity linking. The top-$k$ sequences will then be fed into task-specific downstream modules to produce the desired outputs.

## 3    EXPERIMENTS

**Datasets**    Our experiments focus on two datasets: *HotpotQA* and *Multi-evidence FEVER*. HotpotQA (Yang et al., 2018) includes 113k multi-hop questions. Unlike other multi-hop QA datasets (Zhang et al., 2018; Talmor & Berant, 2018; Welbl et al., 2018), where the information sources of the answers are knowledge bases, HotpotQA uses documents in Wikipedia. Thus, its questions are not restricted by the fixed KB schema and can cover more diverse topics. Each question in HotpotQA is also provided with ground truth support passages, which enables us to evaluate the intermediate retrieval performance. Multi-evidence FEVER includes 20k claims from the FEVER (Thorne et al., 2018) fact verification dataset, where the claims can only be verified using multiple documents. We use this dataset to validate the general applicability of our method.

**Implementation Details**    All the experiments are conducted on a machine with 8 32GB V100 GPUs. Our code is based on Huggingface Transformers (Wolf et al., 2019). Our best retrieval results are predicted using the exact inner product search index (IndexFlatIP) in FAISS (Johnson et al., 2017).

Table 1: Retrieval performance in recall at $k$ retrieved passages and precision/recall/$F_1$.

| Method | HotpotQA | | | FEVER | | |
|---|---|---|---|---|---|---|
| | R@2 | R@10 | R@20 | Precision | Recall | $F_1$ |
| TF-IDF | 10.3 | 29.1 | 36.8 | 14.9 | 28.2 | 19.5 |
| TF-IDF + Linked | 17.3 | 50.0 | 62.7 | 18.6 | 35.8 | 24.5 |
| DrKIT | 38.3 | 67.2 | 71.0 | - | - | - |
| Entity Linking | - | - | - | 30.6 | 53.8 | 39.0 |
| MDR | **65.9** | **77.5** | **80.2** | **45.7** | **69.1** | **55.0** |

Both datasets assume 2 hops, so we fix $n = 2$ for all experiments. Since HotpotQA does not provide the order of the passage sequences, as a heuristic, we consider the passage that includes the answer span as the final passage. [2] In §3.1.3, we show that the order of the passages is important for effective retriever training. The hyperparameters can be found in Appendix B.1.

## 3.1 EXPERIMENTS: RETRIEVAL

We evaluate our multi-hop dense retriever (MDR) in two different use cases: *direct* and *reranking*, where the former outputs the top-$k$ results directly using the retriever scores and the latter applies a task-specific reranking model to the initial results from MDR.

### 3.1.1 DIRECT

We first compare MDR with several efficient retrieval methods that can directly find the top-$k$ passage sequences from a large corpus, including TF-IDF, TF-IDF + Linked, DrKIT and Entity Linking. **TF-IDF** is the standard term-matching baseline, while **TF-IDF + Linked** is a straightforward extension that also extracts the hyperlinked passages from TF-IDF passages, and then reranks both TF-IDF and hyperlinked passages with BM25 [3] scores. **DrKIT** (Dhingra et al., 2020) is a recently proposed dense retrieval approach, which builds a entity-level (mentions of entities) dense index for retrieval. It relies on hyperlinks to extract entity mentions and prunes the search space with a binary mask that restricts the next hop to using hyperlinked entities. On FEVER, we additionally consider an entity linking baseline (Hanselowski et al., 2018) that is commonly used in existing fact verification pipelines. This baseline first uses a constituency parser to extract potential entity mentions in the fact claim and then uses the MediaWiki API to search documents with titles that match the mentions.

Table 1 shows the performance of different retrieval methods. On HotpotQA the metric is recall at the top $k$ paragraphs[4], while on FEVER the metrics are precision, recall and $F_1$ in order to be consistent with previous results. On both datasets, MDR substantially outperforms all baselines.

### 3.1.2 RERANKING

*Reranking* documents returned by efficient retrieval methods with a more sophisticated model is a common strategy for improving retrieval quality. For instance, state-of-the-art multi-hop QA systems usually augment traditional IR techniques with large pretrained language models to select a more compact but precise passage set. On HotpotQA, we test the effectiveness of MDR after a simple BERT-based reranking: each of the top $k$ passage sequences from MDR is first prepended with the original question and then fed into a BERT-like encoder that predicts relevant scores. We train this reranking model with a binary cross-entropy loss, with the target being whether the passage sequence cover both groundtruth passages. We empirically compare our approach with two other existing reranking-based retrieval methods: **Semantic Retrieval** (Nie et al., 2019) uses BERT at both passage-level and sentence-level to select context from the initial TF-IDF and hyperlinked passages; **Graph Recurrent Retriever** (Asai et al., 2020) learns to recursively select the best passage sequence on top of a hyperlinked passage graph, where each passage node is encoded with BERT.

Table 2 shows the reranking results. Following Asai et al. (2020), we use *Answer Recall* and *Support Passage Exact Match (SP EM)* [5] as the evaluation metrics. Even without reranking, MDR is already better than Semantic Retrieval, which requires around 50 BERT encoding (where each encoding involves cross-attention over a concatenated question-passage pair). After we rerank the

---

[2]If the answer span is in both, the one that has its title mentioned in the other passage is treated as the second.

[3]https://pypi.org/project/rank-bm25

[4]As the sequence length is 2 for HotpotQA, we pick the top $k/2$ sequences predicted by MDR.

[5]Whether the final predicted sequence covers both gold passages.

Table 2: HotpotQA reranked retrieval results (input passages for final answer prediction).

| Method | SP EM | Ans Recall |
|---|---|---|
| Semantic Retrieval | 63.9 | 77.9 |
| Graph Rec Retriever | 75.7 | 87.5 |
| MDR (direct) | 65.9 | 75.4 |
| MDR (reranking) | **81.2** | **88.2** |

Table 3: Retriever Model Ablation on HotpotQA retrieval. *Single-hop* here is equivalent to the DPR method (Karpukhin et al., 2020).

| Retriever variants | R@2 | R@10 | R@20 |
|---|---|---|---|
| Full Retrieval Model | 65.9 | 77.5 | 80.2 |
| - w/o linked negatives | 64.6 | 76.8 | 79.6 |
| - w/o memory bank | 63.7 | 74.2 | 77.2 |
| - w/o shared encoder | 59.9 | 70.6 | 73.1 |
| - w/o order | 17.6 | 55.6 | 62.3 |
| Single-hop | 25.2 | 45.4 | 52.1 |

top-100 sequences from the dense retriever, our passage recall is better than the state-of-the-art Graph Recurrent Retriever, which uses BERT to process more than 500 passages. We do not compare the reranked results on FEVER, as most FEVER systems directly use BERT encoder to select the top evidence *sentences* from the retrieved documents, instead of the reranking the documents.

### 3.1.3   ANALYSIS

To understand the strengths and weaknesses of MDR, we conduct further analysis on HotpotQA dev.

**Retrieval Error Analysis**   HotpotQA contains two question categories: *bridge* questions in which an intermediate entity is missing and needs to be retrieved before inferring the answer; and *comparison* questions where two entities are mentioned simultaneously and compared in some way. In Figure 2, we show the retrieval performance of both question types. The case of *comparison* questions proves easier, since both entities needed for retrieval are present in the question.

This case appears almost solved, confirming recent work demonstrating that dense retrieval is very effective at entity linking (Wu et al., 2019).

For the case of *bridge* questions, we manually inspect 50 randomly sampled erroneous examples after reranking. We find that in half of these cases, our retrieval model predicts an alternative passage sequence that is also valid (see Appendix A.1 for examples).

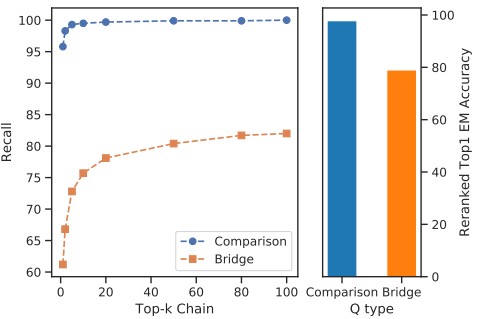

Figure 2: The retrieval performance gap between comparison and bridge questions. Left: recall of groundtruth passage sequences without reranking. Right: Top-1 chain exact match after reranking.

This gives an estimated top-1 passage sequence accuracy of about 90%. Other remaining errors are due to the dense method's inability to capture the exact n-gram match between the question and passages. This is a known issue (Lee et al., 2019; Karpukhin et al., 2020) of dense retrieval methods when dealing with questions that have high lexical overlap with the passages. To this end, a hybrid multi-hop retrieval method with both term and dense index might be used to further improve the performance on *bridge* questions.

**Retriever Ablation Study**   In Table 3, we examine our model with different variations on HotpotQA to show the effectiveness of each proposed component. We see that further training with a memory bank results in modest gains, while using a shared encoder is crucial for the best performance. Respecting the ordering of passages in two hops is essential - training in an order-agnostic manner hardly works at all, and underperforms even the single-hop baseline. Finally, not using hyperlinked paragraphs from TF-IDF passages as additional negatives has only a minor impact on performance.

**Question Decomposition for Retrieval**   As multi-hop questions have more complex structures than simple questions, recent studies (Min et al., 2019; Perez et al., 2020) propose to use explicit question decomposition to simplify the problem. Wolfson et al. (2020) shows that with TF-IDF, using decomposed questions improves the retrieval results. We investigate whether the conclusion still holds with stronger dense retrieval methods. We use the human-annotated ques-

tion decomposition from the QDMR dataset (Wolfson et al., 2020) for analysis. For a question like `Q:Mick Carter is the landlord of a public house located at what address?`, QDMR provides two subquestions, `SubQ1: What is the public house that Mick Carter is the landlord of?` and `SubQ2: What is the address that #1 is located at?`. We sample 100 bridge questions and replace `#1` in `SubQ2` with the correct answer (The Queen Victoria) to `SubQ1`. Note that this gives advantages to the decomposed method as we ignore any intermediate errors. We estimate the performance of potential decomposed methods with the state-of-the-art single-hop dense retrieval model (Karpukhin et al., 2020).

As shown in Table 4, we did not observe any strong improvements from explicit question decomposition, which is contrary to the findings by Wolfson et al. (2020) when using term-based IR methods. Moreover, as shown in the third row of the table, when the 1st hop of the decomposed retrieval (i.e., `SubQ1`) is replaced with the original question, no performance degradation is observed. This suggests that strong pretrained encoders can effectively learn to select necessary information from the multi-hop question at each

Table 4: Comparison with decomposed dense retrieval which uses oracle question decomposition (test on 100 bridge questions). See text for details about the decomposed settings.

| Method | R@2 | R@10 | R@20 |
|---|---|---|---|
| MDR | 54.9 | 63.7 | 70.6 |
| Decomp (SubQ1;SubQ2) | 50.0 | 64.7 | 67.6 |
| Decomp (Q;SubQ2) | 51.0 | 64.7 | 68.6 |

retrieval step. Regarding the performance drop when using explicit compositions, we hypothesize that it is because some information in one decomposed subquestion could be useful for the other retrieval hop. Examples supporting this hypothesis can be found in Appendix A.2. While this could potentially be addressed by a different style of decomposition, our analysis suggests that decomposition approaches might be sub-optimal in the context of dense retrieval with strong pretrained encoders.

## 3.2 EXPERIMENTS: HOTPOTQA

We evaluate how the better retrieval results of MDR improve multi-hop question answering in this section. As our retriever system is agnostic to downstream models, we test two categories of answer prediction architectures: the *extractive* span prediction models based on pretrained masked language models, such as BERT (Devlin et al., 2019) and ELECTRA (Clark et al., 2020), and the retrieval-augmented *generative* reader models (Lewis et al., 2020b; Izacard & Grave, 2020), which are based on pretrained sequence-to-sequence (seq2seq) models such as BART (Lewis et al., 2020a) and T5 (Raffel et al., 2019). Note that compared to more complicated graph reasoning models (Fang et al., 2019; Zhao et al., 2020), these two classes of models do not rely on hyperlinks and can be applied to any text.

**Extractive** reader models learn to predict an answer span from the concatenation of the question and passage sequence ($[q, p_1, ..., p_n]$). On top of the token representations produced by pretrained models, we add two prediction heads to predict the start and end position of the answer span.[6] To predict the supporting sentences, we add another prediction head and predict a binary label at each sentence start. For simplicity, the same encoder is also responsible for reranking the top $k$ passage sequences. The reranking detail has been discussed in §3.1.2. Our best reader model is based on ELECTRA (Clark et al., 2020), which has achieved the best single-model performance on the standard SQuAD (Rajpurkar et al., 2018) benchmark. Additionally, we also report the performance of BERT-large with whole word masking (BERT-wwm) to fairly compare with Asai et al. (2020).

**Generative** models, such as RAG (Lewis et al., 2020b) and FiD (Izacard & Grave, 2020), are based on pretrained seq2seq models. These methods finetune pretrained models with the concatenated questions and retrieved documents as inputs, and answer tokens as outputs. This generative paradigm has shown state-of-the-art performance on single-hop open-domain QA tasks. Specifically, FiD first uses the T5 encoder to process each retrieved passage sequence independently and then uses the decoder to perform attention over the representations of all input tokens while generating answers. RAG is built on the smaller BART model. Instead of only tuning the seq2seq model, it also jointly train the question encoder of the dense retriever. We modified it to allow multi-hop retrieval.

---

[6]To account for yes/no questions, we prepend *yes* and *no* tokens to the context.

More details about these two classes of reader models are described in Appendix B.2.

### 3.2.1 RESULTS

Table 5: HotpotQA-fullwiki test results.

| Methods | Answer | | Support | | Joint | |
|---|---|---|---|---|---|---|
| | EM | F1 | EM | F1 | EM | F1 |
| GoldEn Retriever (Qi et al., 2019) | 37.9 | 48.6 | 30.7 | 64,2 | 18.9 | 39.1 |
| Semantic Retrieval (Nie et al., 2019) | 46.5 | 58.8 | 39.9 | 71.5 | 26.6 | 49.2 |
| Transformer-XH (Zhao et al., 2020) | 51.6 | 64.1 | 40.9 | 71.4 | 26.1 | 51.3 |
| HGN (Fang et al., 2019) | 56.7 | 69.2 | 50.0 | 76.4 | 35.6 | 59.9 |
| DrKIT (Dhingra et al., 2020) | 42.1 | 51.7 | 37.1 | 59.8 | 24.7 | 42.9 |
| Graph Recurrent Retriever (Asai et al., 2020) | 60.0 | 73.0 | 49.1 | 76.4 | 35.4 | 61.2 |
| MDR (ELECTRA Reader) | **62.3** | **75.3** | **57.5** | **80.9** | **41.8** | **66.6** |

**Comparison with Existing Systems**    Table 5 compares the HotpotQA test performance of our best ELECTRA reader with recently published systems, using the numbers from the official leaderboard, which measure answer and supporting sentence exact match (EM)/F1 and joint EM/F1.  Among these methods, only GoldEn Retriever (Qi et al., 2019) does not exploit hyperlinks.  In particular, Graph Recurrent Retriever trains a graph traversal model for chain retrieval; TransformerXH (Zhao et al., 2020) and HGN (Fang et al., 2019) explicitly encode the hyperlink graph structure within their answer prediction models.  In fact, this particular inductive bias provides a perhaps unreasonably strong advantage in the specific context of HotpotQA, which by construction guarantees ground-truth passage sequences to follow hyperlinks. Despite not using such prior knowledge, our model outperforms all previous systems by large margins, especially on supporting fact prediction, which benefits more directly from better retrieval.

**Reader Model Variants**    Results for reader model variants are shown in Table 6.[7] First, we see that the BERT-wwm reader is 1-2% worse than the ELECTRA reader when using enough passages.  However, it still outperforms the results in (Asai et al., 2020) which also uses BERT-wwm for answer prediction.  While RAG and FiD have shown strong improvements over extractive models on single-hop datasets such as NaturalQuestions (Kwiatkowski

Table 6: Reader comparison on HotpotQA dev set.

| | Model | Top k | EM | F1 |
|---|---|---|---|---|
| Extractive | ELECTRA | Top 50 | 61.7 | 74.3 |
| | ELECTRA | Top 250 | 63.4 | 76.2 |
| | BERT-wwm | Top 250 | 61.5 | 74.7 |
| Generative | Multi-hop RAG | Top 4*4 | 51.2 | 63.9 |
| | FiD | Top 50 | 61.7 | 73.1 |

et al., 2019), they do not show an advantage in the multi-hop case.  Despite having twice as many parameters as ELECTRA, FiD fails to outperform it using the same amount of context (top 50).  In contrast, on NaturalQuestions, FiD is 4 points better than a similar extractive reader when using the top 100 passages in both.[8]  We hypothesize that the improved performance on single-hop questions is due to the ability of larger pretrained models to more effectively memorize single-hop knowledge about real-world entities.[9]  Compared to multi-hop questions that involve multiple relations and missing entities, simple questions usually only ask about a certain property of an entity.  It is likely that such simple entity-centric information is explicitly mentioned by a single text piece in the pretraining corpus, while the evidence for multihop questions is typically dispersed, making the complete reasoning chain nontrivial to memorize.  More analysis on RAG can be found in Appendix A.3.

---

[7]For the compute-heavy generative models, we feed in as many passages as possible without running into memory issues (Muli-hop RAG takes top 4 passages from hop1, and for each of those, takes another top 4 from hop2. They are not necessarily the same as the top 16 passages sequences.). As extractive models encode each passage sequence separately, we can use arbitrary number of input sequences. However, the performance mostly plateaus as we use over 200 input sequences.

[8]We implemented NQ extractive readers with both RoBERTa-large and ELECTRA-large, and RoBERTa-large yielded a better answer EM of 47.3, which is much lower than the 51.4 answer EM achieved by FiD.

[9]As shown by Roberts et al. (2020), a large pretrained seq2seq model can be finetuned to directly decode answers with questions as the only inputs. However, we find that this retrieval-free approach performs poorly on multi-hop questions. See Appendix C for the exact numbers.

Table 7: Multi-Evidence FEVER Fact Verification Results. **Loose-Multi** represents the subset that requires multiple evidence *sentences*. **Strict-Multi** is a subset of **Loose-Multi** that require multiple evidence sentences from different *documents*.

| Method | Loose-Multi (1,960) | | Strict-Multi (1,059) | |
|---|---|---|---|---|
| | LA | FEVER | LA | FEVER |
| GEAR | 66.4 | 38.0 | - | - |
| GAT | 66.1 | 38.2 | - | - |
| KGAT with ESIM rerank | 65.9 | 39.2 | 51.5 | 7.7 |
| KGAT with BERT rerank | 65.9 | 40.1 | 51.0 | 6.2 |
| Ours + KGAT with BERT rerank | **77.9** | **42.0** | **72.1** | **16.2** |

**Inference Efficiency** To compare with existing multi-hop QA systems in terms of efficiency, we follow Dhingra et al. (2020) and measure the inference time with 16 CPU cores and batch size 1. We implement our system with a fast approximate nearest neighbor search method, *i.e.*, HNSW (Malkov & Yashunin, 2018), which achieves nearly the same performance as exact search. With an in-memory index, we observe that the retrieval time is negligible compared to the forward pass of large pretrained models. Similarly, for systems that use term-based indices, the BERT calls for passage reranking cause the main efficiency bottleneck. Thus, for systems that do not release the end-to-end code, we estimate the running time based on the number of BERT cross-attention forward passes (the same estimation strategy used by Dhingra et al. (2020)), and ignore the overhead caused by ad-

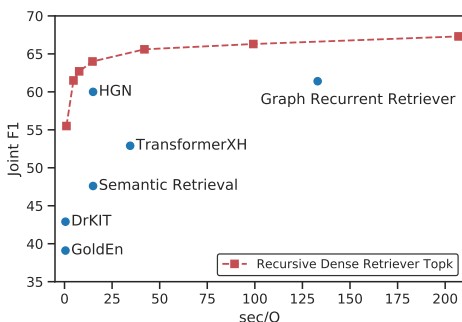

Figure 3: Efficiency-performance trade-off comparison with published HotpotQA systems. The curve is plotted with different number of top $k$ ($k$=1,5,10,20,50,100,200) passage sequences we feed into the reader model. seq/Q denotes the time required for each query.

ditional processing such as TF-IDF or linking graph construction. As shown in Figure 3, our method is about 10 times faster than current state-of-the-art systems while achieving a similar level of performance. Compared to two efficient systems (DrKIT and GoldEn), we achieve over 10 points improvement while only using the top-1 retrieval result for answer and supporting sentence prediction.

## 3.3 EXPERIMENTS: MULTI-EVIDENCE FEVER

For FEVER claim verification, we reuse the best open-sourced verification system, *i.e.*, KGAT (Liu et al., 2020), to show the benefit of our retrieval approach over existing retrieval methods. We report the results in verification *label accuracy* (LA) and the *FEVER score*[10] in Table 7, where the numbers of competitive baselines, GEAR (Zhou et al., 2019), graph attention network (GAT) (Veličković et al., 2017) and variants of KGAT are from the KGAT (Liu et al., 2020) paper. All these baselines use entity linking for document retrieval, then rerank the sentences of the retrieved documents, and finally use different graph attention mechanisms over the fully-connected sentence graph to predict verification labels. Since some instances in the multi-evidence subset used by previous studies only needs multiple evidence *sentences* from the same document, we additionally test on a strict multi-hop subset with instances that need multiple *documents*. As shown by the results, even without finetuning the downstream modules, simply replacing the retrieval component with MDR leads to significant improvements, especially on the strict multi-evidence subset.

## 4 RELATED WORK

**Open-domain QA with Dense Retrieval** In contrast to sparse term-index IR methods that are widely used by existing open-domain QA systems (Chen et al., 2017; Wang et al., 2018; Yang et al., 2019), recent systems (Lee et al., 2019; Guu et al., 2020; Karpukhin et al., 2020) typically

---

[10]FEVER scores takes into account both support sentence accuracy and label accuracy, similar as the joint metrics in HotpotQA.

uses dense passage retrieval techniques that better capture the semantic matching beyond simple n-gram overlaps. To generate powerful dense question and passage representations, these methods either conduct large-scale pretraining with self-supervised tasks that are close to the underlying question-passage matching in retrieval, or directly use the human-labeled question-passage pairs to finetune pretrained masked language models. On single-hop information-seeking QA datasets such as NaturalQuestions (Kwiatkowski et al., 2019) or WebQuestions (Berant et al., 2013), these dense methods have achieved significant improvements over traditional IR methods. Prior to these methods based on pretrained models, Das et al. (2019) use RNN encoder to get dense representations of questions and passages. They also consider an iterative retrieval process and reformulate the query representation based on reader model's hidden states. However, their method requires an initial round of TF-IDF/BM25 retrieval and a sophisticated RL-based training paradigm to work well. Finally, like the aforementioned methods, only single-hop datasets are considered in their experiments. More akin to our approach, Feldman & El-Yaniv (2019) use a similar recursive dense retrieval formulation for multi-hop QA. In contrast to their biattenional reformulation component applied on top of token query and passage representations, we adopt a more straightforward query reformulation strategy, by simply concatenating the original query and previous retrieval as the inputs to the query encoder. Together with stronger pretrained encoders and more effective training methods (in-batch + memory bank negative sampling vs their binary ranking loss), MDR is able to double the accuracy of their system.

**Query Expansion Techniques in IR**    As our dense encoder augments the original question with the initial retrieved results to form the updated query representation, our work is also relevant to query expansion techniques (Rocchio, 1971; Voorhees, 1994; Ruthven & Lalmas, 2003) that are widely used in traditional IR systems. In particular, our system is similar in spirit to pseudo-relevance feedback techniques (Croft & Harper, 1979; Cao et al., 2008; Lv & Zhai, 2010), where no additional user interaction is required at the query reformulation stage. Existing studies mainly focus on alleviating the uncertainty of the user query (Collins-Thompson & Callan, 2007) by adding relevant terms from the first round of retrieval, where the retrieval target remains the same throughout the iterative process. In contrast, the query reformulation in our approach aims to follow the multi-hop reasoning chain and effectively retrieves different targets at each step. Furthermore, instead of explicitly selecting terms to expand the query, we simply concatenate the whole passage and rely on the pretrained encoder to choose useful information from the last retrieved passage.

**Other Multi-hop QA Work**    Apart from HotpotQA, other multi-hop QA datasets (Welbl et al., 2018; Talmor & Berant, 2018; Zhang et al., 2018) are mostly built from knowledge bases (KBs). Compared to questions in HotpotQA, questions in these datasets are rather synthetic and less diverse. As multi-hop relations in KBs could be mentioned together in a single text piece, these datasets are not designed for an open-domain setting which necessitates multi-hop retrieval. Existing methods on these datasets either retrieve passages from a small passage pool pruned based on the the specific dataset (Sun et al., 2019; Dhingra et al., 2020), or focus on a non-retrieval setting where a compact documents set is already given (De Cao et al., 2018; Zhong et al., 2019; Tu et al., 2019; Beltagy et al., 2020). Compared to these research, our work aims at building an efficient multi-hop retrieval model that easily scales to large real-world corpora that include millions of open-domain documents.

## 5    CONCLUSION

In this work, we generalized the recently proposed successful dense retrieval methods by extending them to the multi-hop setting. This allowed us to handle complex multi-hop queries with much better accuracy and efficiency than the previous best methods. We demonstrated the versatility of our approach by applying it to two different tasks, using a variety of downstream modules. In addition, the simplicity of the framework and the fact that it does not depend on a corpus-dependent graph structure opens the possibility of applying such multi-hop retrieval methods more easily and broadly cross different domains and settings.

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

# A QUALITATIVE ANALYSIS

## A.1 FALSE BRIDGE QUESTION ERROR CASES

As mentioned in §3.1.3, half of the errors of bridge questions are not real errors. In Table 8, we can see that the model predicts alternative passage sequences that could also be used to answer the questions.

Table 8: Error cases where our model predicts a passage sequence that is also correct. Important clues are marked in blue.

**Q:** What languages did the son of Sacagawea speak?
**Ground-truth SP Passage Titles:** Charbonneau, Oregon; Jean Baptiste Charbonneau
**Predicted:**
1. Museum of Human Beings: Museum of Human Beings, included in the National American Indian Heritage Month Booklist, November 2012 and 2013 is a novel written by Colin Sargent, which delves into the heart-rending life of Jean-Baptiste Charbonneau, the son of Sacagawea. Sacagawea was the Native American guide, who at 16 led the Lewis and Clark expedition.
2. Jean Baptiste Charbonneau: Jean Baptiste Charbonneau (February 11, 1805 – May 16, 1866) was an American Indian explorer, guide, fur trapper trader, military scout during the Mexican-American War, "alcalde" (mayor) of Mission San Luis Rey de Francia and a gold prospector and hotel operator in Northern California. He spoke French and English, and learned German and Spanish during his six years in Europe from 1823 to 1829. He spoke Shoshone, his mother tongue, and other western American Indian languages...

**Q:** Altnahinch is located in a county that has a population density of how many per square mile?
**Ground-truth SP Passage Titles:** Altnahinch Dam; County Antrim
**Predicted:**
1. Altnahinch: Altnahinch is a townland in County Antrim, Northern Ireland.
2. County Antrim: County Antrim (named after the town of Antrim, from Irish: "Aontroim", meaning "lone ridge" , )) is one of six counties that form Northern Ireland. Adjoined to the north-east shore of Lough Neagh, the county covers an area of 3046 km2 and has a population of about 618,000. County Antrim has a population density of 203 people per square kilometer / 526 people per square mile...

**Q:** What foundation do scholars give for the likelihood of collaboration on a William Shakespeare Play written between 1588 and 1593?
**Ground-truth SP Passage Titles:**
Authorship of Titus Andronicus, William Shakespeare's collaborations
**Predicted:**
1. Titus Andronicus: Titus Andronicus is a tragedy by William Shakespeare, believed to have been written between 1588 and 1593, probably in collaboration with George Peele. It is thought to be Shakespeare's first tragedy, and is often seen as his attempt to emulate the violent and bloody revenge plays of his contemporaries, which were extremely popular with audiences throughout the 16th century.
2. William Shakespeare's collaborations: Like most playwrights of his period, William Shakespeare did not always write alone... Some of the following attributions, such as "The Two Noble Kinsmen", have well-attested contemporary documentation; others, such as "Titus Andronicus", are dependent on linguistic analysis by modern scholars...

**Q:** Zach Parise's father played in which league?
**Ground-truth SP Passage Titles:** Jordan Parise; Zach Parise
**Predicted:**
1. Zach Parise: Zachary Justin Parise (born July 28, 1984) is an American professional ice hockey left winger who is currently serving as an alternate captain for the Minnesota Wild in the National Hockey League (NHL). He has also played for the New Jersey Devils, where he served as team captain and led the team to the 2012 Stanley Cup Finals. Parise's father, J. P. Parisé...
2. J. P. Parisé: Jean-Paul Joseph-Louis Parisé (December 11, 1941 – January 7, 2015) was a Canadian professional ice hockey coach and player. Parise played in the National Hockey League (NHL), most notably for the Minnesota North Stars and the New York Islanders.

## A.2 EXAMPLES FROM THE QUESTION DECOMPOSITION ANALYSIS

Table 9: Sampled retrieval errors (marked in red) *only* made by the decomposed system. These errors could be potentially avoided if the model has access to the full information in the original question or previous hop results. The important clue for correctly retrieving the documents or avoiding errors is marked in blue. Once decomposed, the marked information are not longer available in one of the decomposed retrieval hop.

---

**Multi-hop Question**: What is the birthday of the author of "She Walks These Hills"?
**Decomposed Questions**:
1. Who is the author of She Walks These Hills?
2. What is the birthday of Sharyn McCrumb?
**Ground-truth SP Passages**:
She Walks These Hills: She Walks These Hills is a book written by Sharyn McCrumb and published by Charles Scribner's Sons in 1994, which later went on to win the Anthony Award for Best Novel in 1995.
Sharyn McCrumb: Sharyn McCrumb (born February 26, 1948) is an American writer whose books celebrate the history and folklore of Appalachia. McCrumb is the winner of numerous literary awards...
**Decomposed Error Case:**
1. She Walks These Hills (✓)
2. Tané McClure: Tané M. McClure (born June 8, 1958) is an American singer and actress.

---

**Multi-hop Question:** When was the album with the song Unbelievable by American rapper The Notorious B.I.G released?
**Decomposed Questions:**
1. What is the album with the song Unbelievable by American rapper The Notorious B.I.G?
2. When was the album Ready to Die released?
**Ground-truth SP Passages:**
Unbelievable (The Notorious B.I.G. song): Unbelievable is a song by American rapper The Notorious B.I.G., recorded for his debut studio album Ready to Die...
Ready to Die: Ready to Die is the debut studio album by American rapper The Notorious B.I.G.; it was released on September 13, 1994, by Bad Boy Records and Arista Records...
**Decomposed Error Case:**
1. Unbelievable (The Notorious B.I.G. song) (✓)
2. Ready to Die (The Stooges album): Ready to Die is the fifth and final studio album by American rock band Iggy and the Stooges. The album was released on April 30, 2013...

---

**Multi-hop Question:** Whose death dramatized in a stage play helped end the death penalty in Australia?
**Decomposed Questions:**
1. What is the stage play that helped end the death penalty in Australia?
2. Whose death was dramatized in Remember Ronald Ryan?
**Ground-truth SP Passages**:
Barry Dickins: Barry Dickins (born 1949) is a prolific Australian playwright, author, artist, actor, educator and journalist... His most well-known work is the award winning stage play "Remember Ronald Ryan", a dramatization of the life and subsequent death of Ronald Ryan, the last man executed in Australia...
Ronald Ryan: Ronald Joseph Ryan (21 February 1925 – 3 February 1967) was the last person to be legally executed in Australia. Ryan was found guilty of shooting and killing warder George Hodson during an escape from Pentridge Prison, Victoria, in 1965...
**Decomposed Error Case:**
1. Capital punishment in Australia: Capital punishment in Australia has been abolished in all jurisdictions. Queensland abolished the death penalty in 1922. Tasmania did the same in 1968, the federal government abolished the death penalty in 1973, with application also in the Australian Capital Territory and the Northern Territory...
2.Ronald Ryan(✓)

---

### A.3 EXTRACTIVE & GENERATIVE READER MODEL

Table 6 demonstrates the answer prediction performance for four different reader models. The extractive models predict answers given the top 250 retrieved passage sequences (pairs of passage from hop1 and hop2). Since generative models are generally heavier on the computation side, we can only use fewer passages. Besides the observations alredy discussed in §3.2.1, we

Table 10: Answer EM using top 50 retrieved passage chains

| Model | Overall | Comp (20%) | Bridge (80%) |
|---|---|---|---|
| ELECTRA | 61.7 | 79.0 | 57.4 |
| FiD | 61.7 | 75.3 | 58.3 |

hypothesize the worse performance of multi-hop RAG compared to FiD is partially due to the smaller pretrained model used in RAG, i.e., BART is only half the size of T5-large. Also, as RAG back-propagate the gradients to the query encoder, it needs more memory footprint and can only take in fewer retrieved contexts. Our RAG implementation largely follows the implementation of the original paper and we did not use the PyTorch checkpoint (as used by FiD) to trade computation for memory. We conjecture the multi-hop RAG performance will also improve if we augment the current implementation with memory-saving tricks. However, given the same amount of context and read model size, the multi-hop RAG is still worse than the extractive ELECTRA reader, i.e., with only the top 1 retrieved passage sequence, our ELECTRA reader gets 53.8 EM compared to the 51.2 answer EM achieved by multi-hop RAG when using more context.

Given the same number of retrieved passage sequences (top 50) as shown in table 10, FiD obtains similar performance to ELECTRA, despite that the generative model can generate arbitrary answers for the given input. (We tried constrained decoding for the generative model. However, no significant performance improvements were observed, indicating that the errors from the generative model are not due to the free-form generation task.) Further question type analysis in HotpotQA showed that the main difference comes from the comparison type of question, while for bridge question, FiD performs slightly better than ELECTRA. This finding might indicate that for generation models, numerical comparison is still a bigger issue compared to extractive models.

## B MODEL DETAILS

### B.1 BEST MODEL HYPERPARAMETERS

Table 11: Hyperparameters of Retriever

| | |
|---|---|
| learning rate | 2e-5 |
| batch size | 150 |
| maximum passage length | 300 |
| maximum query length at initial hop | 70 |
| maximum query length at 2nd hop | 350 |
| warmup ratio | 0.1 |
| gradient clipping norm | 2.0 |
| traininig epoch | 50 |
| weight decay | 0 |

Table 12: Hyperparameters of Extractive Reader (ELECTRA)

| | |
|---|---|
| learning rate | 5e-5 |
| batch size | 128 |
| maximum sequence length | 512 |
| maximum answer length | 30 |
| warmup ratio | 0.1 |
| gradient clipping norm | 2.0 |
| traininig epoch | 7 |
| weight decay | 0 |
| # of negative context per question | 5 |
| weight of SP sentence prediction loss | 0.025 |

## B.2 FURTHER DETAILS ABOUT READER MODELS

### B.2.1 EXTRACTIVE READER

The extractive reader is trained with four loss functions. With the `[CLS]` token, we predict a reranking score based on whether the passage sequence match the groundtruth supporting passages. On top of the representation of each token, we predict a answer start score and answer end score. Finally, we prepend each sentence with the `[unused0]` special token and predict whether the sentence is one of the supporting sentences using the representations of the special token. At training time, we pair each question with 1 groundtruth passage sequence and 5 negative passage sequence which do not contain the answer. At inference time, we feed in the top 250 passage sequences from MDR. We rank the predicted answer for each sequence with a linear combination of the reranking score and the answer span score. The combination weight is selected based on the dev results.

### B.2.2 FUSION-IN-DECODER

The FiD model uses T5-large as the underlying seq2seq model. It is twice as large as the extractive models and has 770M parameters. We reuse the hyperparameters as described in Izacard & Grave (2020). The original FiD uses the top 100 passages for NaturalQuestions. In our case, we use the top 50 retrieved passage sequences and concatenate the passages in each sequence before feeding into T5. In order to fit this model into GPU, we make use of PyTorch checkpoint [11] for training.

### B.2.3 MULTI-HOP RAG

The RAG model aims to generate answer $y$ given question $x$ and the retrieved documents $z$. Similarly, the goal of multi-hop RAG can be expressed as: generate answer $y$ given question $x$ and retrieved documents in hop one $z_1$ and hop two $z_2$ (Limiting to two hops for HotpotQA). The model has three components:

- Hop-one retriever $p_{\eta_1}(z_1|x)$ with parameter $\eta_1$ to represent the retrieved top-k passage distribution (top-k truncated distribution) given the input question $x$.

- Hop-two retriever $p_{\eta_2}(z_2|x, z_1)$ with parameter $\eta_2$ to represent the hop-two retrieved top-k passage distribution given not only the question $x$ but also the retrieved document $z_1$ from hop-one.

- A generator $p_{\theta}(y_i|x, z_1, z_2, , y_{1:i-1})$ to represent the next token distribution given input question $x$, hop-one retrieved document $z_1$, hop-two retrieved document $z_2$ and previous predicted token $y_{1:i-1}$ parametrized by $\theta$

**Multi-Hop RAG Sequence Model**     As the RAG Sequence model, this model generates the answer sequence given the fixed set of documents from hop-one retriever and hop-two retriever. In order to the get the probability of the generated sequence, we marginalize through the two latent variables corresponding to the two retrieval hops:

$$p_{sequence}(y|x) =$$

$$\sum_{z_1} p_{\eta_1}(z_1|x) \sum_{z_2} p_{\eta_2}(z_2|x, z_1) \prod_{i}^{N} p_{\theta}(y_i|x, z_1, z_2, y_{1:i-1})$$

$$\sum_{z_1} \sum_{z_2} p_{\eta_1}(z_1|x) p_{\eta_2}(z_2|x, z_1) \prod_{i}^{N} p_{\theta}(y_i|x, z_1, z_2, y_{1:i-1})$$

where $z_1$ and $z_2$ are top k document from the respective retrieval modules.

---

[11] https://pytorch.org/docs/stable/checkpoint.html

**Multi-Hop RAG Token Model**   Moreover, the model can make predictions based on different passage extracted at each token.

$$p_{token}(y|x) =$$
$$\prod_i^N \sum_{z_1} \sum_{z_2} p_{\eta_1}(z_1|x) p_{\eta_2}(z_2|x, z_1) p_\theta(y_i|x, z_1, z_2, y_{1:i-1})$$

The predicted probability for each token is the following

$$p_{token}(y_i|(x, y_j)) =$$
$$\sum_{z_1} \sum_{z_2} p_{\eta_1}(z_1|x) p_{\eta_2}(z_2|x, z_1) p_\theta(y_i|x, z_1, z_2, y_{1:i-1})$$

## C   RETRIEVAL-FREE APPROACHES

Inspired by a recent work (Roberts et al., 2020) that trains the T5 seq2seq model to directly decode answers from questions (*retrieval-free*), we conduct similar experiments on HotpotQA using BART (Lewis et al., 2020a). As shown in Figure 4, the performance gap between retrieval-based methods and retrieval-free methods on multi-hop QA is much larger than the gap in the case of simple single-hop questions.

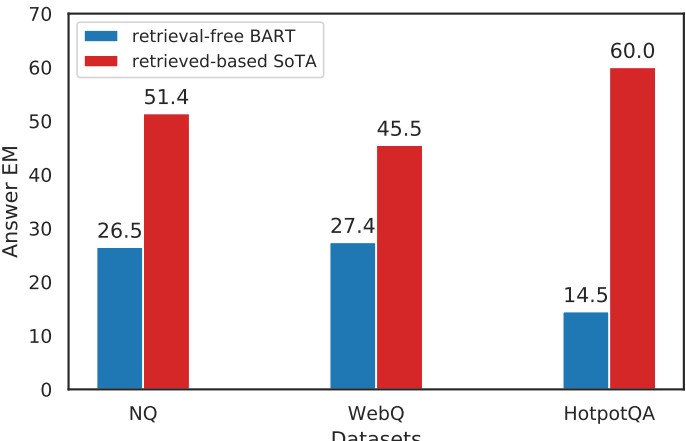

Figure 4: Performance gap between retrieval-free and retrieval-based methods on different QA datasets.

## D   A UNIFIED QA RETRIEVAL SYSTEM

In practice, when a fixed text corpus is given for open-domain systems, we do not know beforehand whether the incoming questions require single or multiple text evidence. Thus, it is essential to build a unified system that adaptively retrieves for multiple hops. Due to the simplicity of the approach, our method can easily be extended in the unified setup. To the best of our knowledge, only (Asai et al., 2020) test the same retrieval method on both single and multi-hop questions but with separate trained models. Here we take a further step and explore the possibility of using a single retrieval model for both types of questions.

To enable adaptive retrieval, we add a binary prediction head on top of the question encoder. Once the retriever finishes the 1-hop retrieval, it encodes concatenation of $q$ and $p_1$ and predicts whether to stop retrieval using the final hidden state of the first token. We construct this unified setting with NaturalQuestions-Open (Lee et al., 2019) (NQ) as single-hop and HotpotQA as multi-hop. As the two datasets use different corpora, we merge the two[12] for easy comparison. As baselines,

---

[12]The Wikipedia corpus of NQ is taken from DPR (Karpukhin et al., 2020).

we use the retrieval models trained only on the respective dataset. For HotpotQA, the baseline is the best multi-hop retrieval model discussed in the main text. For NQ, we follow the training method in DPR (Karpukhin et al., 2020), but with a shared question and passage encoder, which achieves stronger results. As the NQ corpus includes multiple passages of the same document and the HotpotQA corpus only uses the introduction passage, we are not able to compute the strict title-based support passage recall for HotpotQA as in §3.2. Thus, we only evaluate answer recall. Results are in Table 13. In contrast to existing studies that train different models for each dataset, we show that a unified dense retrieval model can maintain competitive performance on both, despite the vastly different nature of both datasets. Note that the information-seeking questions in NQ is usually noisier and more ambiguous, while HotpotQA questions are more complicated and contains more lexical overlaps with the evidence passages. Specifically, for NQ, the unified retrieval model achieves very similar performance as the single-dataset DPR model, while the performance on HotpotQA decreases more. We conjecture that this is because the information-seeking questions in NQ cover more diverse patterns, and the added HotpotQA training questions do not cause a dramatic distribution shift from the NQ test data. We leave the development of a more general retrieval system that handles different styles of questions to future work.

Table 13: Comparing the unified retrieval model with models specifically trained for each task. We test the retrieval performance with a single merged corpus. For easy comparison, all three models are based on BERT-base encoder which we find achieves stronger performance than RoBERTa-base on NQ. AR@K denotes answer recall at top-K retrieved passage sequences.

| Model | NQ | | HotpotQA | |
|---|---|---|---|---|
| | AR@20 | AR@100 | AR@20 | AR@100 |
| single-hop only | 80.7 | 87.3 | - | - |
| multi-hop only | - | - | 83.4 | 89.4 |
| unified | 79.5 | 86.1 | 78.1 | 83.0 |

