# OpenReview forum: "Answering Complex Open-Domain Questions with Multi-Hop Dense Retrieval"
_ICLR.cc/2021/Conference — ICLR 2021 Poster_

### Official Review · AnonReviewer1 · 2020-10-28
**This paper provides several significant findings that are expected to be referred to by many other studies. Their method is simple and outperforms other multi-hop QA models. Also, it is computationally efficient.**

**Rating:** 9
**Confidence:** 4

**Review:**

1. Summary of this paper
    - The topic of this paper is multi-hop QA, which studies answering complex natural language questions. Complex questions require an information aggregation process across multiple documents and recent multi-hop QA models design this process by sequentially retrieving relevant documents (Asai 2020 et al.). This paper alleviates two problems in recent multi-hop QA models. One is that recent multi-hop QA models require external knowledge such as Wikipedia hyperlinks. This problem results in the models' low generalization ability on new domains that the external knowledge is no longer available. The other problem is computational efficiency. The authors propose a novel multi-hop QA model named MDR that does not require external knowledge and is ten times faster than the recent models. MDR uses question reformulation and MIPS. Question reformulation design the information aggregation process by iteratively generating a query vector related to the documents that should be accompanied to answer the original question. MDR generates such query vectors by comparing the given question and previously retrieved documents. MDR encodes passages in a large corpus(indexing) with the same encoder used in the question reformulation process and uses MIPS to find relevant documents with the generated query vectors. In experiments, the authors show that MDR outperforms recent multi-hop QA models, and also they show the computational efficiency of MDR.

2. Strong and weak points of this paper
    - Strong points
        - This paper provides a detailed analysis of their method. Experimental results show the validity of the proposed method, and some strong findings described below.
            - Table 2 confirms that MDR outperforms "Graph Rec Retriever (Asai et al.)". This result shows the feasibility of a more accurate multi-hop QA model without external knowledge such as Wikipedia hyperlinks.
            - Table 3 shows a detailed analysis of each component in MDR. This table indicates several vital features for multi-hop QA models that can be easily ignored in the model design process. The experimental results on "w/o order" and "w/o linked negatives" show significant findings in multi-hop QA.
            - Table 4 shows that the question reformulation method (MDR) has similar performance to the question decomposition method with human-annotated sub-questions.
            - Table 5 shows the end-to-end performance of multi-hop QA models. MDR outperforms existing state-of-the-art multi-hop QA models.
        - The proposed method is computationally efficient.
        - The proposed method is simple. Many follow-up studies based on the proposed method are expected.
        - The experimental results support their claim.
    - Weak points
        - This paper does not mention the publicly available code of their method. It would be nice if the authors provide implementations after the decision process.
        - In the section "Question Decomposition for Retrieval," the authors conclude that question decomposition is unnecessary in the context of dense retrieval with a strong pretrained encoder. However, Table 4 shows that question decomposition with a simple open-domain QA model has a similar performance to MDR. These results indicate that question decomposition is an effective method to make simple single-hop open-domain QA models used in multi-hop QA. Please provide more evidence for the conclusion, "unnecessity of question decomposition."

3. Recommendation
    - Accept
    - This paper provides several significant findings that are expected to be referred to by many other studies. Their method is simple and outperforms other multi-hop QA models. Also, it is computationally efficient.

4. Questions
    - In Table 4, the Decomp method is based on DPR (dense passage retriever). What will be the results if MDR uses the gold sub-questions? Does using sub-questions in MDR increase retriever performance?
    - Please provide the number of hard negative samples for a question.
    - In section 2.2, what is the start token in the sentence " Specifically, we apply layer normalization over the start token’s representations from RoBERTa to get the final dense query/passage vectors." Is the start token pooled_output of the CLS token or hidden representation of the CLS token?

---

> ### Author Response · Authors · 2020-11-20
> **Authors' reply to R1**
>
> We thank R1 for recognizing our efforts in proposing this simple and efficient model. **We will release publicly available code and models upon paper acceptance.**
>
> **On Question Decomposition for Retrieval:** It is worth mentioning that the decomposition-based performance in Table 4 is based on manually labeled gold decompositions (details in the last paragraph of Sec 3.1.3). For the convenience of analysis, we ignore the  errors from both any possible decomposition models and also  the answer prediction from earlier hops. In practice, when using a trained  decomposition model and on off-the-shelf single-hop retrieval model, the performance gap will become larger. In addition, collecting training data for question decomposition is also nontrivial. Thus, the “explicit decomposition  + off-the-shelf single-hop retrieval” paradigm used by [1] is less likely to achieve better performance than our simple concatenation-based query reformulation approach.
>
>
> > In Table 4, the Decomp method is based on DPR (dense passage retriever). What will be the results if MDR uses the gold sub-questions? Does using sub-questions in MDR increase retriever performance?
>
> Unfortunately, we don’t have the gold decomposed sub-questions for hotpot QA training data. For this analysis, we manually annotate 100 gold sub-questions based on the BREAK dataset (https://allenai.github.io/Break/). Since MDR is not trained with sub-questions and it only learns latent decomposition, we did not test it with sub-questions.
>
> > Please provide the number of hard negative samples for a question.
>
> The number of hard negative examples for each training question is 2.
>
> > In section 2.2, what is the start token in the sentence " Specifically, we apply layer normalization over the start token’s representations from RoBERTa to get the final dense query/passage vectors." Is the start token pooled_output of the CLS token or hidden representation of the CLS token?
>
> We use the hidden representation of the CLS token.
>
>
> [1] Break it down: A question understanding benchmark. Wolfson et al.

---

### Official Review · AnonReviewer4 · 2020-10-29
**Reasonable Idea, Solid Experiment, Strong Results, Lack of Novelty**

**Rating:** 6
**Confidence:** 5

**Review:**

Summary:
This paper proposes multi-hop dense retrieval for open-domain multi-hop question answering. It extends previous dense passage retrieval into the corresponding multi-hop version by using retrieved passages to latently reformulate the query representation after each retrieval pass. In the end, it can significantly improve the performance on HotpotQA and multi-evidence FEVER dataset. The analyses are very comprehensive and extensive from almost every relevant perspective.

Pros:
1.	Extending dense passage retriever into its multi-hop version is a reasonable direction. This is the first work in this direction.
2.	The experimental results are strong, and the analyses are comprehensive.
Cons:
1.	This paper mainly focuses on the experimental and analysis part. Although it is good to know these lessons in multi-hop dense retrieval, the proposed method itself is limited in terms of novelty.
2.	Since the idea mainly comes from dense passage retrieval, it is not very clear to me whether the improvement comes from the dense passage retrieval or the proposed latent query reformulation for DPR. Since these are the major contribution of this paper, I think it is necessary to have a separate section for discussion.

Question:
1.	In Table 3, what is the training detail of the Single-hop ablation? Is it also trained through a similar negative sampling process described in section 2.2? If so, why the improvement of multi-hop version DPR is so significant?
2.	In section 1, it is said that “The main problem in answering multi-hop open-domain questions is that the search space grows exponentially with each retrieval hop.” In my understanding, the proposed multi-hop DPR still suffers from this problem, right? The only difference between the proposed multi-hop DPR and previous approaches is that it does not use any structured knowledge within the documents for retrieval.

---

> ### Author Response · Authors · 2020-11-20
> **Authors' reply to R4**
>
> We thank R4 for appreciating our experimental results.
>
> **Novelty:** Please see our reply to R2 about the novelty concerns.
>
>
>
> **On disentangling the effect of dense retrieval and query reformulation:**:
>
> > it is not very clear to me whether the improvement comes from the dense passage retrieval or the proposed latent query reformulation for DPR
>
>  In table 4, we have shown the performance of vanilla dense retrieval without the recursive formulation. We did consider some other query reformulation approaches, such as using RNN taking the query and previous passage dense representations as inputs. However, these methods usually fall short compared to simple concatenation. Compared to the latent formulation, it is also possible to consider explicit query reformulation, i.e., generate a new query sequence at each step. This direction shares spirits with explicit query decomposition, as discussed in 3.1.1. According to our experiments, explicitly generating new query tokens might not be necessary given the powerful query encoder.
>
> **Question on the single-hop baseline:**
> > what is the training detail of the Single-hop ablation? Is it also trained through a similar negative sampling process described in section 2.2? If so, why the improvement of multi-hop version DPR is so significant?
>
> The negative sampling process of this baseline is similar to the original DPR. Here we consider all the supporting passages as positive passages. The gap is mainly due to the nature of multi-hop questions. By design, these questions usually do not include sufficient clues to retrieve passages in further hops. Thus, even if we train the single-hop model to retrieve all supporting passages, it can usually only retrieve the initial hop at inference time.
>
> **Question on the search space challenge:**:
>
> > In my understanding, the proposed multi-hop DPR still suffers from this problem, right? The only difference between the proposed multi-hop DPR and previous approaches is that it does not use any structured knowledge within the documents for retrieval.
>
> The exponentially growing search space is a challenge for all multi-hop QA systems. Previous solutions tackle this challenge by simply looking at the Wikipedia hyperlinks. This is actually a strong inductive bias as the HotpotQA is constructed with hyperlinks. Our method does not get away with the search space problem; however, we are tackling this challenge in a more general way. We show that dense retrieval can efficiently handle the search space even without the help of corpus-specific structures.

---

### Official Review · AnonReviewer3 · 2020-10-29
**A simple and effective pseudo relevance feedback approach to multi-hop QA**

**Rating:** 7
**Confidence:** 4

**Review:**

Summary; The paper proposes a simple, clever, and as far as I can tell novel, combination of dense retrieval techniques and pseudo relevance feedback for multi-hop (complex) open-domain QA. The basic idea is to concatenate the passages returned for the first query to the original question, to form a new query to be encoded and used in combination with the retrieval system. Thus, the paper does not present a radically new idea but combines successfully recent popular methods (dense retrieval, featuring in most SOTA work in QA) with old techniques well known and studied for decades in the IR community.

In general, I enjoyed reading this paper and find this is a useful technique that the community should learn about to explore further in the same and other related problems; e.g., OpenQA. The approach is somewhat incremental, but competitive and conceptually simple. The lack of awareness for the IR context is my main reason for the score not being higher.

Pros:
- The paper shows that the idea is effective in terms of performance, yielding state-of-the-art results on two multi-hop datasets, HotpotQA and multi-evidence FEVER.
- Experiments seem generally rigorous and reproducible following standard popular datasets and procedures.
- Experiments include very recent work on reader’s architectures, including generative ones (RAG, FID).
- The architecture’s simplicity does not make assumptions on the nature of the data and associated meta information (e.g., link graphs) and produces also a more efficient system.

Cons:
- The paper could be organized better in its final version. In particular, providing more context and motivation for the problem; first of all, why are such “complex” questions important? How key is the multi-hopness aspect? I would particularly recommend anchoring this discussion to the findings of (Min et al, 2019, https://www.aclweb.org/anthology/P19-1416/). In particular the fact that many such questions can be solved in one hop. How natural/artificial is the task? How does this aspects affects this specific study?
- The related work section is insufficient and the absence of an adequate discussion of the (pseudo) relevance feedback work in IR is a major weakness. This is an foundational line of work going back at least to the research of Rocchio in the 1970s. I would suggest (Ruthven and Lalmas, 2003, A survey on the use of relevance feedback for information access systems) as a starting point.

Detailed feedback/comments:
- How does the use of tfidf as a source of hard negatives relate to the argument about IR baselines being poor?
- What is the motivation for, and conclusion for, evaluating additional linked docs as negatives if they only yield minor gains?
- Could you discuss more the nature of the supervised information available with respect to the fact that the number of hops is known, and also the order of the passage sequences (or can be inferred heuristically); e.g., wrt to the claim “training in an order-agnostic manner hardly works at all, and underperforms even the single-hop baseline”.
- Have you tried using more than 2 steps? It may be valuable to run an experiment.
- It would be also valuable to experiment with this approach on other OpenQA tasks such as Natural Question etc. Also to provide more evidence for the generality of the approach.
- How does this approach deal with the limited encoder capacity, in terms of number of tokens? How many passages can you append to form the 2nd query, and how does this affect performance?

---

> ### Author Response · Authors · 2020-11-20
> **Authors' reply to R3 (1/2): related work and importance of the problem**
>
> We thank R3 for the detailed review and the pointer on related IR work.
>
> **Related IR work**: We omitted the discussion about related IR techniques in the original version mainly due to space constraints. We have now added another paragraph discussion query expansion techniques in the related work section. Briefly, our approach is relevant to the existing pseudo-relevance feedback techniques, in the sense that we augment the original query with the context in the initial retrieval, and both our approach and pseudo-relevance feedback do not require additional human feedback for the query augmentation step. The differences are: 1. In our multi-hop task setting, the retrieval targets vary at each step. 2. We do not explicit select terms to append the original query. Instead, we rely on the pretrained models to select useful info from the concatenated query and passage.
>
> **On the necessity of multi-hop reasoning (Min et al., 2019):** We are aware of the line of work questioning the multi-hop nature of HotpotQA ([1], [2], [3]). However, these studies mainly focus on a less realistic distractor setting where no retrieval is required. In the open-domain setting, the “shortcuts” phenomenon is usually alleviated because there are much more distracting passages. As shown by Min et al. (Section 3.2), their single-hop model fails to achieve competitive performance. Our ablation study 3.1.3 also confirms the necessity of multi-hop retrieval for this task.
>
> **On the importance of the multi-hop QA problem:**: There are two primary motivations for studying the multi-hop QA problem: First, it is an important research direction towards developing NLU models that go beyond one-shot semantic matching and have the reasoning ability to aggregate multiple evidence pieces. Besides, building multi-hop QA models is also important in practice as it can further improve the coverage of QA systems and answer those long-tail questions to which the answers are not explicitly stated by a single piece of text in the knowledge source. We have updated the paper with more motivations in the Introduction section.
>
> [1] Understanding Dataset Design Choices for Multi-hop Reasoning. Chen et al., 2019
> [2] Compositional Questions Do Not Necessitate Multi-hop Reasoning. Min et al., 2019
> [3] Avoiding Reasoning Shortcuts: Adversarial Evaluation, Training, and Model Development for Multi-Hop QA. Jiang et al., 2019

---

> ### Author Response · Authors · 2020-11-20
> **Authors' reply to R3 (2/2): Questions**
>
> Please see below our responses to the raised questions:
>
> > How does the use of tf-idf as a source of hard negatives relate to the argument about IR baselines being poor
>
> Although the tf-idf baseline is much worse than the proposed model, the tf-idf retrieved passages are still more relevant than in-batch passages. Thus, those passages can provide better negative signals to train the retrieval model. Using tf-idf is mainly inspired by the existing DPR approach [4]. However, it is also possible to use an existing dense retriever to collect hard negatives, as shown by a concurrent study [5].
>
> >  the motivation for, and conclusion for, evaluating additional linked docs as negatives
>
> The motivation for using those linked passages as additional negatives is that tf-idf usually cannot retrieve those passages that are relevant to the 2nd hop because the important entity (bridge entity) for the 2nd hop retrieval is missing in the original question. Thus, initially, we conjectured that if we only use tf-idf negatives, then we won’t have strong training signals to learn the 2nd hop retrieval. However, our later experiments showed that those linked negatives are not significant for the final performance.
>
> > the order of the passage sequences ... wrt to the claim “training in an order-agnostic manner hardly works at all, and underperforms even the single-hop baseline
>
> Concerning the order information, it is reasonable to train the model to follow the faithful reasoning order that is understandable to humans. Given that the original question does not include the necessary information for further hops, training the model to retrieve the 2nd hop without using the 1st hop information might force the model to learn superficial matching patterns, which might not generalize. Further, if we train the model using randomly shuffled reasoning chains, then the correlation between the training queries and retrieval targets might be too corrupted for the model to effectively learn the “natural” reasoning process, resulting in even worse performance than single-hop baseline (Table 3).
>
> > Have you tried using more than 2 steps
>
> We haven’t tried using more than two hops for the considered datasets. Mainly because these datasets always only provide 2-hop supervision. While some existing studies have tried to insert other passages to the gold 2 hop reasoning chain to create multiple training chains, for simplicity, we did not use those data augmentation techniques. However, given training samples, which include arbitrary length of reasoning hops, it is possible to add a simple prediction head on top of the encoder to predict whether to stop retrieving (as discussed in our reply to the next question).
>
> > other OpenQA tasks such as Natural Question
>
> We did try our model on the NQ dataset. However, since the dataset is guaranteed to be single-hop on Wikipedia, we observe neither significant performance improvements nor degeneration when conducting two hops retrieval on NQ. Perhaps more interestingly, we also considered a unified setting combining NQ and HotpotQA and trained the model to retrieve additional hops adaptively. As shown in Appendix D (updated version), a trained unified retrieval model can achieve competitive performance on both datasets. To the best of our knowledge, most existing studies treat these datasets as separate tasks and train specific models for each.
>
> >  How does this approach deal with the limited encoder capacity, in terms of number of tokens? How many passages can you append to form the 2nd query, and how does this affect performance?
>
> The maximum number of tokens is limited by the maximum context length allowed by the pretrained model (512). We always only append one document to form the 2nd query (we maintain K augmented queries for beam search) in this work. The effect of appending multiple passages might be exploited in future work, given the recent advances of pretrained models that more efficiently encode long context.
>
> [4] Dense Passage Retrieval for Open-Domain Question Answering. Kaipukhin et al, 2020
> [5] Approximate Nearest Neighbor Negative Contrastive Learning for Dense Text Retrieval  https://openreview.net/forum?id=zeFrfgyZln

---

### Official Review · AnonReviewer2 · 2020-10-31
**Official Blind Review #2**

**Rating:** 5
**Confidence:** 3

**Review:**

This paper extends the recently proposed dense retrieval methods to the multi-hop open-domain questions, so as to handle complex multi-hop queries. The overall idea is simple, direct but effective. The authors conduct extensive experiments on two multi-hop datasets, HotpotQA and multi-evidence FEVER, and evaluation results demonstrate that the proposed model achieves impressive results on both the knowledge retrieval task and multi-hop QA.

My only concern is about the novelty of the paper. The contribution of this paper seems to be limited as it just combines the recently proposed dense retrieval methods with multi-hop QA. Besides, compared with [1], this paper seems to just replace the RNN-based encoder in the knowledge retriever with the BERT-based encoder.

[1] Das R, Dhuliawala S, Zaheer M, et al. Multi-step retriever-reader interaction for scalable open-domain question answering[C].ICLR, 2019.

---

> ### Author Response · Authors · 2020-11-20
> **Authors' reply to R2**
>
> Thanks R2 for the review and for pointing us to the related work.
>
> **Novelty:**
> We understand R1’s concern about novelty given the simplicity of our formulation and model architecture. Due to the exciting challenges posed by the multi-hop QA datasets, we have already seen a line of sophisticated and carefully designed approaches (e.g., injecting inductive bias of the Wikipedia hyperlinked graph into model architectures) achieving strong performance on the particular HotpotQA dataset. While we appreciate how the existing models effectively leverage the semi structures of the corpus, we are more interested in finding a general solution for the multi-hop QA task itself, instead of modeling the corpus-specific structures in the particular dataset. The proposed simple approach mainly come from this goal in mind, and we highlight the main contribution as follows:
>
> * **We are the first to show the challenging multi-hop QA task can be effectively solved by a much simpler method, without exploiting the dataset-specific hyperlinks.** As our approach is based on existing dense retrieval methods, it might look straightforward in hindsight, however, it is worth noting that existing dense retrieval methods mostly only show superior performance in scenarios where semantic matching beyond lexical overlaps is essential (e.g., dense retrieval is not able to outperform bm25 on SQuAD). The HotpotQA task actually shares some characteristics with SQuAD, as it involves lots of lexical overlap between the queries and passages. This could be one of the potential concerns which prevent previous work exploring dense retrieval in this space. Our experiments clear out this concern and first show that dense retrieval is useful for this task. (We conjecture this is mainly due to the dense retriever’s ability to effectively link the bridge entities.) Compared to the complex pipelines used by existing SOTA systems, we believe our simple and general framework will give researchers new insights about the multi-hop retrieval problem.
> * **Better accuracy-efficiency trade-off**: Thanks to the simplicity of our method, we can achieve much better efficiency compared to existing sophisticated SOTA methods.
> * **Improvements on the dense retrieval itself**: On top of the standard bi-encoder dense retrieval methods, we have proposed to use a *shared-encoder* and the *memory-bank mechanism* to further improve the performance.
> * **Thorough analysis on both the retrieval and answer prediction tasks for multi-hop QA**:  In Sec. 3.1.3,  we discussed the performance on different question types and an alternative solution that uses explicit question decomposition. In Sec, 3.2.1, we compare generative and extractive answer prediction modules and observe different phenomena compared to single-hop open-domain questions. We believe these are new and valuable insights about multi-hop QA.
>
> **Related work (Das et al., 2019):** While the iterative formulation in Das et al. is relevant to our method (discussion added in related work of the revision), there are some key differences:
> * First of all, the QA tasks studied by our work and Das et al. are different. Das et al. did not study the task of multi-hop QA. They focus on the case of single-hop QA, where the retrieval targets of their iterative retriever remain the same at each step, i.e., finding the passage that includes the answer. For multi-hop QA, the retrieval targets at each step are different, and our model is trained to infer a “chain of reasoning” which is faithful and sufficient to answer the multi-hop question.
> * In order to outperform the baseline systems, Das et al. (Table 1 and Section 5 in their paper) need an initial sparse retrieval stage to form a smaller candidate pool. When directly applied to a corpus with 1.6M passages, their system could not match the previous baseline performance. In contrast, our system does not need a BM25/TF-IDF component and can be directly applied to a large corpus with 5M passages. While being much simpler than previous pipelined systems, our method still achieves stronger performance.
> * The query reformation stage of Das et al. requires access to the hidden states of a pretrained reader model and uses the reader model to assign rewards to train the retrieval model with RL. We do not assume any access to the states of an answer extraction. In fact, as the retrieval target of multi-hop QA at each retrieval step is not necessarily the final answer passage, the signal from the answer extraction model might not be as useful as in the single-hop QA case. As our retrieval model directly operates on the 5M corpus, we refrain from using RL due to its poor sample efficiency, given a large action space.

---

### Author Response · Authors · 2020-11-20
**General response to reviewers and the new revision**

We thank all reviewers for their thoughtful feedback. We are glad that the reviewers appreciate the simplicity/effectiveness of our proposed approach (R1,2,3), and find our experiments to be solid/rigorous/extensive (R2, 3, 4). Please see our detailed response below and check the updated paper which reflects the reviewers comments about related IR work and problem motivations.

---

### Decision · Program_Chairs · 2021-01-07
**Final Decision**

**Decision:**

Accept (Poster)

**Comment:**

The paper introduces improving passage retrieval for multi-hop QA datasets by recursively retrieving passages, adding previously retrieved passages to the input (in addition to a query). This simple method shows gains on multiple QA benchmark datasets, and the evaluation presented in the paper on multiple competitive benchmark datasets (HotpotQA, FEVER) is very thorough (R1, R3, R4).

While the application is pretty narrow, the performance gain (considering both efficiency and accuracy) is fairly significant, and the paper presents a simple model with less assumption (e.g., inter-document hyperlinks), that could be useful for future research.

[1] also seems like a relevant line of work.

[1] Generation-Augmented Retrieval for Open-domain Question Answering
https://arxiv.org/pdf/2009.08553.pdf